# Pavement Quality Index Rating Strategy Using Fracture Energy Analysis for Implementing Smart Road Infrastructure

**DOI:** 10.3390/s21124231

**Published:** 2021-06-20

**Authors:** Samuel Abejide, Mohamed M. H. Mostafa, Dillip Das, Bankole Awuzie, Mujib Rahman

**Affiliations:** 1Civil Engineering Department, Faculty of Science and Engineering Technology, Ibika Campus, Walter Sisulu University, Eastern Cape 4960, South Africa; 2Sustainable Transportation Research Group (STRg), Civil Engineering Department, College of Agriculture, Engineering and Science, Howard College Campus, University of Kwa-Zulu Natal, KwaZulu-Natal 4041, South Africa; mostafam@ukzn.ac.za (M.M.H.M.); dasd@ukzn.ac.za (D.D.); 3Centre for Sustainable Smart Cities, Department of Built Environment, Faculty of Engineering, Built Environment and Information Technology, Willows, Central University of Technology, Bloemfontein 9300, South Africa; bawuzie@cut.ac.za; 4Civil Engineering Department, School of Engineering and Technology, Birmingham, Aston University, Birmingham B4 7ET, UK; m.rahman19@aston.ac.uk

**Keywords:** pavement durability, smart roads infrastructure, pavement quality index, pavement management system

## Abstract

Developing a responsive pavement-management infrastructure system is of paramount importance, accentuated by the quest for sustainability through adoption of the Road Traffic Management System. Technological advances have been witnessed in developed countries concerning the development of smart, sustainable transportation infrastructure. However, the same cannot be said of developing countries. In this study, the development of a pavement management system at network level was examined to contribute towards a framework for evaluating a Pavement Quality Index and service life capacity. Environmental surface response models in the form of temperature and moisture variations within the pavement were applied, using sensor devices connected to a data cloud system to carry out mathematical analysis using a distinctive mesh analysis deformation model. The results indicated variation in the Resilient Modulus of the pavement, with increasing moisture content. Increase in moisture propagation increased saturation of the unbound granular base which reduced the elastic modulus of the sub-base and base layer and reduced the strength of the pavement, resulting in bottom-up cracks and cracking failure. The horizontal deformation reduced, indicating that the material was experiencing work hardening and further stress would not result in significant damage. Increasing temperature gradient resulted in reduced stiffness of the asphalt layer. In tropical regions, this can result in rutting failure which, over time, results in top-down cracks and potholes, coupled with increasing moisture content.

## 1. Introduction

Infrastructure has been described as an important enabler of economic growth and national competitiveness [1]. Sustainable transportation, on the other hand, refers to the ability of a transport system to be fully maintained and operated at minimal cost with optimal efficiency. The attainment of this scenario depends on the budget allocation for maintenance plans as well as resources (qualified engineers, traffic control and regulation systems as well as favourable and controlled environment and climatic conditions). The advent and growth of transportation infrastructure networks in recent times have increased the demand for an efficient management plan to be developed by the agencies in charge of them with regards to their maintenance and operation. In many situations, this activity has not received the required attention owing to resource scarcity for asset management operations [2]. This, in turn, contributes to the economic losses that are a consequence of the processes of infrastructure deterioration and their cost of repair [2].

Understanding the impact of transportation resulting from methodological applications with regards to the related paradox in social and distributional impact is of primary importance [3,4]. Bassiago [4] reiterated that, for sustainable development to be attained, any approach would require economic planning that attempts to foster economic growth. A major consequence faced by countries is when they spend in excess of what is necessary for a project of lower priority. This will result in a negative effect of lower productivity in the drive towards efficiency and managing fund allocation for pavement maintenance.

Pavement challenges in many countries stem from various factors which range from environmental, climatic, and human factors, to functional factors pertaining to design-service loading, and construction defects amongst many others [5,6]. The methods of how these challenges are addressed will vary depending on the extent of damage as well as the trigger factor. Insufficient funding for maintenance results in most of the challenges not being addressed within the time of detection (crack initiation) up until the time when full deformation becomes visible, which causes discomfort in driving and riding quality. Hence, there is a need for a smart approach to monitoring the structural health of pavements.

### 1.1. Network Level Pavement Management System

The structural condition of pavements can be considered from two main viewpoints: that of the road user and that of the road engineer. In general, road users view the condition of the pavement on the basis of its performance requirements. These usually affect the quality of travel, notably comfort, safety, and operating costs. On the other hand, the road engineer views the pavement on the basis of the functional requirement of the road as it relates to the load-bearing capacity as well as the stiffness requirement to be maintained during the design service-life, at optimal cost of operation. Depending on the data and information obtained about the performance level of the pavement in question, the necessary treatment or maintenance remedy to be adopted would be considered by the roads engineer.

One main method involved in pavement management systems is that of crack sealing/filling. This method is one of the most widely used treatments in the world because of its low cost and relatively fast application rate. The essence of this management system is to prevent water ingress through the cracks, generated as a result of pavement distress during the service life, seeping through to the base and underlying inter-pavement layers [7,8,9,10].

This Pavement Management System, as a concept of micro-surfacing, developed in 1970 in Germany and used in the United States since 1980, involves the mixture of polymer-modifying asphalt emulsion, water, mineral filler, and other additives, being spread over the pavement surface. Its use has increased significantly because of the flexibility of its application requirement. It has no restriction on traffic volume, and only needs one hour to cure before it can carry traffic again [8,11].

On the other hand, this pavement management system at network level has resulted in the development of methods to analyse pavement distress. Some network methods include: the Artificial Neural Network (ANN), which is regarded as a very useful tool all over the world, especially for complex datasets for which manual function designs are unsatisfactory; and the Monte-Carlo approach, which involves the process of creating partitions and breaking the entire pavement into segments to be analysed discretely. In the sampling process, a uniform distribution is used. However, in order to facilitate a robust and semi-automated analysis, the data are usually stored in a three-dimensional (3D) array.

Depending on the type of data collected, the method of data processing and analysis will differ because of the several data sources and data formats. Data could range from environmental/climatic data, traffic data, axle-load data, to pavement-geometry data parameters. All the stored data collected are processed and analysed using the Finite Element approach in MATLAB, ABAQUS, and ANSYS, or improved advancement in Multi-Layered Elastic Pavement analysis. The PMS is a series of activities which involves planning and executing maintenance of a road pavement aimed at minimising budget while maximising performance and extended life span. To achieve this, an understanding of the current behaviour of the pavement condition is necessary through assessment.

This is done by computational analysis based on pavement performance indicators, classified as: Mode I Failure (crack opening and initiation resulting from the formation of stresses at the top of the pavement), and Mode II Failure (increasing shear stress acting normal to the surface of the pavement resulting in the formation of longitudinal cracks, transverse cracks, rutting, fatigue and pavement ride discomfort). Since International Roughness Index (IRI) is not a distress value, and therefore, it is not listed in the (American Society for Testing and Materials) ASTM D6433-10, the points to be deducted during the analysis can be computed using correlation points established between the Pavement Quality Index and International Roughness Index [11,12,13].

Marshall, Meier and Welch [14] developed an improved pavement instrumentation and non-destructive testing platform for testing daily and seasonal environmental effect on pavements. This attempt resulted in the development of a data logging system for pavement sub-grade and asphalt-concrete layer, taking into account the adverse impact of environmental factors over time. Zuo [15] made use of an instrumentation test of four pavement sections by installing a weather station at the test site to monitor air temperature, precipitation, relative humidity, wind speed, and solar radiation after every 60 s, with an hourly total and average being logged. The pavement response was recorded and validated using Falling Weight Deflectometer (FWD) test readings on the pavement stress-strain measurements. This method requires the use of a FWD vehicle which contains a load plate and sensors. FWD method enables departments of transportation to back-calculate the modulus of elasticity of pavement layers and determine the structural integrity of pavements in the network. This method gives an approximate idea of the structural behaviour of the pavement of an asphalt-concrete-wearing surface [5].

In the above method, pavement performance distresses are obtained from still, visual images taken by vans as they pass over the network, and typically involves using third-party software which calculates certain distresses on the surface layer as prescribed by the department or user. This enables departments of transportation, road asset management agencies, etc. to understand where there are greater densities of distresses in the network. Data capturing technologies for recording these images have increased dramatically in recent years with high resolution and three-dimensional images becoming available [16].

Recent advancements in mobile data collection technology, such as Street Inventory Management Systems (SIMS) and Computer Aided Dispatch Systems (CADS), have led to increased departmental use of automated distress surveys [17]. The implementation of automated processes enables any department of transportation (specifically the Eastern Cape Department of Transportation, South Africa) to detect distresses as well as other pavement characteristics (deflection, and sub-grade stress and strain) at a network level, which requires advanced data collection and analysis of pavement distress, using mathematical models and numerical analysis.

Many methods to capture pavement performance values exist [5,15], for example, the assessment of pavement surface distress by visual observation, measurement of pavement surface roughness by using Roughometer-II, deflection measurements by using the Benkelman Beam Method, or traffic volume surveys over 24-h. These methods range from using mathematical models to methods involving the use of an automated data collection system. The first stage, when using an automated data collection system, involves driving a van at a constant speed over a network of roads to calculate and obtain the roughness of the pavement. Based on how the suspension of the vehicle behaves with respect to the road, a certain roughness value is calculated and stored at very short increments [18].

With regard to the ever-increasing technological trend and developmental strategy in the world, advancements in technology and Big Data as they relate to the internet of things (IoT), however negligible, result in increasing pressure on countries to improve the performance of infrastructure systems (transportation facilities, health centres, appropriate building infrastructure management, etc.) that incorporate activities from the onset of conceptualisation and planning, through implementation, to construction and maintenance. In order to attain efficiency, it is necessary to modify the current Pavement Management System, being employed for pavement maintenance and rehabilitation, with respect to delayed response time. Some of the pitfalls (delayed pavement investigation, increasing cost overrun, time constraints, feedback response time, etc.) associated with the conventional PMS, include analysis of pavement condition surveys using Excel Sheets, the Microsoft SQL server, and backlog data which are analysed after a delay, owing to the robust database analysis system. Methods to implement a PMS are usually influenced by political concerns, budget, and time to mobilise to the site of the failed section [19,20].

In order to establish a Pavement Management System that is robust and efficient, it is necessary to identify measures to monitor the conditions of exposure during the pavement’s life regarding distress modes, environmental conditions, design methods, construction methods as well as necessary maintenance plans to be implemented [6]. The applicability, and subjection of, a given pavement section to Pavement Management System (PMS) models, such as Genetic Algorithm, Monte Carlo Simulation, Evolutionary Algorithm, and Artificial Neural Networks ANN have been examined in the past with positive results. The limitations of such models include: long computational time, premature convergence, limited capacity to fine-tune results, robustness of the pavement sections in terms of total length of roads to be analysed, and errors resulting from complexity (non-homogenous material property) of pavement sections to be analysed [15].

### 1.2. Overview of Pavement Instrumentation

Prior to the early 1920s, pavement design was based purely on experience [5,15]. The same thickness of road section was used regardless of the type of sub-grade soils. The empirical methods of previous pavement design have been improved to take into account the effect of sub-grade soils using soil classification methods and strength tests, mainly the California Bearing Ratio (CBR) test. In recent years, the notion of how to build roads has gone from a fully empirical approach to the use of semi-automated analytical methods [21,22].

Electrical resistance strain gauges are usually used to measure strain in flexible pavement (a highly elastic pavement layer of asphalt-concrete). For this reason, strain gauges are selected according to their gauge length. For pavement application, the working principle is based on the maximum aggregate size of the pavement mixture matrix, usually taken to be 25.4 mm. The criterion for maximum aggregate size method was developed by [23], which relates the level of flexibility resulting from movement of the aggregate molecules within the pavement structure *(meso structure)* [23]. The maximum aggregate size takes account of the parameters of the multi-phase properties considered within the pavement material such as shape, size, gradation, and distribution of the aggregate particles. The interaction or friction developed between the maximum aggregate size and the bituminous binder defines the resultant strain developed within the pavement structure [23,24,25]. This change in length, or deflection caused results into induced strain variations occurring within the pavement structure measures using strain gauge. Some devices used to measure asphalt-concrete strain are: H-Gauges and Strip Gauges; Foil strain gauges cemented to, or embedded in, carrier blocks prepared in the laboratory; and Soil Strain Gauges. One of the most commonly used strip H-gauges is manufactured by Kyowa (Kyowa Gauges); see Figure 1. This gauge (strip) consists of a 120 Ω, unbonded, metallic-filament (wire) strain gauge embedded in acryl with a modulus of elasticity of 2800 MPa (400,000 psi). Aluminum, steel, or brass anchor bars may be used at the ends of the strip. At the Technical University of Denmark, H-gauges have been modified to improve their accuracy and durability [23]. The strain gauge is completely embedded in a strip of fibre glass reinforced with epoxy with low stiffness but high flexibility and strength [24,25].

Some recent devices for measuring asphalt-concrete strain are Transducers. These consist of a bonded, metallic-foil, strain gauge embedded in thin sheets of asphalt mastic to measure longitudinal strain at the bottom of the asphalt-concrete layer. Further research led to the development of pressure cells which measure the electrical output that can be generated from the device with regards to the stress applied to the diaphragm of the cell in a mechanical bench calibration test. The requirements and the design for the pressure cell can be found in Tabatabee et al. [26,27].

Other forms of measuring devices have been noted, which range from velocity devices to detect driver speed, as well as deflection devices: Linear Variable Differential Transformer (LVDT), Multi-Depth Deflectometer (MDD), and Single Layer Deflectometer (SLD), to determine the pavement deflection under moving loads. These forms of measuring make use of static devices which are used periodically and not permanently as they are very sophisticated and fragile. Certain factors affect the reliability of the measured strain and make this approach not sustainable. These factors are: the different positions of the installed gauges; the non-uniformity of the pavement material matrix; the layer thickness of the pavement; and the dynamic load profile resulting from the pavement surface roughness, load cycles, truck suspensions, axle configuration, and tyre-pressure rating [24].

HMA pavement is a composite structure composed of complex, heterogeneous, layered, material components formed by multiple combinations of different materials subjected to traffic loading and various environmental conditions. Asphalt-concrete pavement is one of the most robust infrastructure components in the world, and its design, construction, conservation and rehabilitation depend on the availability of resources, including finance.

However, the prediction of the design service-life of the pavement is one of the most challenging tasks for pavement engineers [19]. Certain factors affecting the performance of asphalt-concrete, namely, the rate of loading, type of axle load, loading time, rest period, temperature mode of loading, and increasing moisture content, result in premature pavement failure. As such, different empirical and mechanistic models and analytical frameworks have been developed in order to predict asphalt-concrete responses in service conditions [5,25,26].

### 1.3. The Purpose and Significance of This Study

The concept of smart pavement-management is currently gaining acceptance as a result of trends in technology advancement emerging during the Fourth Industrial Revolution (4IR), using a Network Level Pavement Management System.

In an Australian report on Smart Infrastructure, it was reiterated that smart infrastructure development should take three imperatives into consideration: “establish a Market, enhance attractiveness of private financing of infrastructure and overhaul infrastructure for radical innovation and productivity growth” [19,20]. As a result of the ever-increasing need for sustainability and sustainable development, as it concerns promoting development without hampering future growth, there needs to be appropriate management of infrastructure assets [16]. Hence, with the implementation of sensor devices to aid in monitoring the structural health of pavements, sustainable development will be achieved.

The technologies mentioned above, used together with the proposed evaluation of smart monitoring of pavement structural health, will enable departments to have very detailed and precise understanding of the condition of the underlying pavement, throughout the network, during its service life.

For the purpose of this study, time domain sensors were installed to collect relevant data from the pavement test section. The results from the analysis of the data indicated a thin line of variation between the seasonal change in flexible pavement response to environmental conditions when compared with the FWD. In order to introduce a real-time data logging system to analyse the variation in responses, it was necessary to set up a system to manage an automated data-base of pavement networks to detect pavement distress.

Therefore, in this paper the development of a Pavement Quality Index rating for Smart Road Infrastructure has been proposed, using smart pavement-sensors to collect and analyse pavement temperature and moisture saturation content to assess the quality and performance of pavements, as well as to predict pavement failure within any specific time-frame in real-time, using constitutive models within the Mechanistic-Empirical Pavement Design Guide synchronised with a Pavement Management System Model [5,6,17].

## 2. Materials and Methods

To achieve an efficient PMS, it is necessary to perform a comprehensive assessment of the road condition continuously in real-time throughout the pavement’s design life. This entails: surface friction, i.e., skid resistance; the structural adequacy of the road, i.e., pavement stiffness; roughness of the road, i.e., riding quality; and surface distress, i.e., loading and environmental conditions. The pavement roughness is obtained as a deduction from Pavement Condition Rating (*PCR*)—this value is a function of the traffic loading resulting from the extent of visible stresses on the test section. However, it is necessary to establish a mathematical model for the International Roughness Index (*IRI*) to validate the judgement made for a Pavement Quality Index (*PQI*) as shown in Equation (1).
(1)$$PQI=PCR-a\;{\left(IRI\right)}^{b}$$
where: *a* = 0.00004915; *b* = 2.4230; *IRI* is a variable based on the road section considered. For new pavements, *IRI* ranges between (0.20–2.0 m/km) and (1.01–3.50 m/km).

For a standard road section, the pavement condition is determined using Equation (2).
(2)$$PCR=100\;{\left(\frac{12-IRI}{12}\right)}^{a}{\left(\frac{D_{max}-DP}{D_{max}}\right)}^{b}$$
where: *IRI* = International Roughness Index, m/km;

*D*_max_ = maximum possible deductible points due to distress;*DP* = Actual total deductible points;*a* and *b* are constants.

Owing to the short-comings of the (American Society for Testing and Materials; West Conshohocken, PA, USA) ASTM D6433-07 Standard Distress Catalogue which is substantial and renders the (Pavement Condition Index) PCI calculation a time-consuming process, several transportation agencies have resorted to simplifying the methodology, mainly with respect to how the distress is identified during pavement surface inspections and how it should be quantified [2,5]. Only a few types of distress are surveyed instead of the totality of situations defined by the ASTM D6433-07 standard [28]. However, it is imperative to understand which distress best represents the pavement’s condition in order to design a simplified procedure for calculating the PCI. To that end, the distresses included in the ASTM D6433-07 Standard Catalogue [28] have been modified for this study, based on the family of defects (distortions, cracking, and weathering) and where they occur (pavement edge or carriageway) in order to carry out a constitutive model within the Mechanistic-Empirical Pavement Design Guide. A Pavement Management System has been proposed that was designed to produce a smart responsive report sheet in real-time, considering moisture and temperature gradient variations synchronised with a web-based pavement deformation architecture model, with real-time, surface response failure prediction mode.

For the purpose of this study, a new approach towards pavement instrumentation has been proposed. This approach made use of constitutive models within the Mechanistic-Empirical Pavement Design Guide synchronised with a Pavement Management System. The response models were collected via a cloud-based, smart surface-response model in a report sheet, in real-time, considering moisture and temperature gradient variations synchronised with a web-based, pavement deformation architecture model.

Figure 2 shows the proposed system for implementing smart road-management infrastructure using moisture and temperature sensors to measure the condition of the road at any given time. The Finite Element Method of analysis and Multi-Layer Elastic Design of pavement are still in use but do not give real-time solutions to determine a Pavement Quality Index. The Arduino block diagram in Figure 2 shows the proposed method for determining the characteristic variables that change over time during the life-cycle of the pavement. The block diagram shows the microprocessor connection between the sensor and the data storage system provided in synchrony with the Amazon Web Service. This study provides the synchrony between the Arduino sensor (temperature and moisture/humidity) system and the AWS algorithm system.

Great leaps in the advancement of transportation systems have made it possible for greater distances to be covered in a unit of time, resulting from efficient and safe riding pavement surfaces [29]. Pavement design and construction are generally carried out to keep the underlying layers unsaturated. However, information for evaluating pavement response while in service is limited. The advent of smart devices for pavement monitoring and management is not entirely a new concept [5,30,31]. Thus, technological improvements are achieved through the introduction of fundamentally new solutions (innovations), and through incremental improvements in existing techniques and systems (product and process innovation). Hence, the proposed smart device for monitoring pavements. The approach adopted for this study was to make use of a technology that can be correlated with pavement design principles [31,32]. A description of the sequence for collecting the response data from the pavement is presented in Figure 3.

### 2.1. Components of Smart Road-Management Infrastructure

The components of a smart road-management system are derived from advancements in Micro-Electro-mechanical Sensors and Systems (MESM). This innovative technology and wireless networking of sensors employs advanced methods in Structural-Health Monitoring (SHM) which is a systemic approach that is employed to monitor and prevent rapid deterioration of infrastructure assets such as dams, bridges, and buildings [6]. However, not much application is seen in real-time data generation on road pavement.

The objective of the smart component for this study (microprocessor; PCB; humidity, temperature, and motion sensors) owed reference to Industry 4.0 technology. The aim was to generate response models which could be analysed further within a complex mathematical environment, either individually or as a combination of all response models, obtained within the medium of exposure or analysis. The response of each failure-reaction model would depend on the geometry of the material and the nature of loading to which it is exposed. One of the most critical functionalities in the analysis of any pavement type is the ability to identify the mode of failure and its growth progression along the line of loading. For Smart Roads Infrastructure, instrumentation of the road pavement with tracking and monitoring devices was proposed in order to read and analyse pavement response to changing environmental and loading conditions remotely. The components of Smart Road Infrastructure sensor devices required some form of GPRS and a web-based application system to be active. The web application was designed to read, compute, and analyse variables associated with environmental conditions, loading cycles as well as material requirements. Whereas such activity is dependent on the information to be obtained, a sensor is usually incorporated into the system, built with a micro-processor to read, compute, and analyse pavement response to a physical situation under specified conditions.

The process was such that the detectable, characteristic behaviour was converted into an output signal via a firmware development/gateway. The signal was passed to other parts of the system, which displayed or recorded the measurement and/or used it for control purposes. Similar to any other man-made structural system, failure could occur in different modes, depending on the service condition to which it is subjected.

Smart Road Infrastructure requires wired sensors that are currently available, in order to track pavement response to environmental and traffic conditions, and to generate readings for traffic load failure, temperature and moisture stress and strain failure, as well as deflection failure. The aim of the use of IoT in pavement response and modelling is to assist with generating web-based application data and provide solutions where necessary [12]. This study was limited to smart road infrastructure using moisture and temperature sensor instrumentation built in Arduino system and in synchrony with computational analysis performed in Amazon Web Service. The authors did not consider machine learning algorithm. This procedure using machine learning algorithms is recommended in further study.

#### Experimental Set-Up of Smart Instrumentation for Pavements

For this study, a pavement section was identified in East London within the Eastern Cape Region of South Africa. The North East Express (*NEX*-*East London*) by-pass expressway was selected. This was based on the increasing traffic on the route that was resulting in damage to the road section. The sensor instrumentation was installed in the main carriage-way of the road, along the tyre path which experiences pavement distress more often than any path along the roadway, based on the standard equivalent axle loading as well as standard tyre pressure and axle-separation spacing for passenger cars, trucks, and single-unit buses [5].

The road section was drilled to the depth of the sub-base, as this is the weaker layer of the road exposed to moisture damage more than any layer. The depth of insertion was 400 mm, based on the layout section of the roadway. The sensor device was inserted and sealed with the base course material to a compaction depth of 150 mm, then covered with base course material and asphalt-concrete-top waste to another depth of 200 mm and, finally, sealed with HMA to a depth of 60 mm. The assumption for the core closure was based on attaining maximum stiffness at the point of instrumentation to prevent underground ponding of the sub-base and base course, which might pose a threat to the pavement while exposed to traffic loading. The sensor instrumentation is shown in Figure 4a,b. Figure 4a shows a top view of the sensor at the instrumentation site. Figure 4b provides detailed view of the sensor embedded into the road before refill and compaction.

### 2.2. Pavement Performance Stiffness Variable Indicators

According to Ahmed et al. [33], accurate prediction of a Pavement Stiffness Indicator is a requirement for effective pavement performance. This has been a difficult indicator to predict with precision. The Mechanistic-Empirical (M-E) Method of pavement analysis has been used widely to model pavement performance and longevity [34]. This method helps in the determination of distress models, mainly fatigue cracking and fatigue rutting. M-E pavement analysis is basically non-linear by virtue of the material characteristics of the unbound, granular, base-layer material. Consequently, such analysis alone does not give a full-scale pavement prediction when computing the non-linear behaviour of the Pavement Performance Index. The development of a Smart Road Infrastructure failure response model that takes into account the input design variables under finite exposure conditions is necessary [31,32,35]. The South African Pavement Engineering Manual (SAPEM) [36] is a reference manual for all aspects of pavement design. SAPEM is a guide to best practice. The South Carolina Department of Transportation adopts new highway pavement design procedures as set forth in the Guide for Mechanistic-Empirical Design of New and Rehabilitated Pavement Structures, Final Report [34]. The design of pavement, considering PQI rating, for this study, introduced a strategic change in the design constitutive models from analogue/manual input-output to real-time design input-output simulation. Equation (3) indicates the magnitude of Resilient Modulus sustained, resulting from environmental conditions of moisture and temperature variations.
(3)$$E_{r}=2555\;\times \;{\left[\frac{{\left(51.712\right)}^{\left(\frac{1}{0.3586\times GWT^{0.1192}}\right)}}{W_{c}}\right]}^{0.64}$$
where: *GWT* = ground-water table, *W_c_* = moisture content which plays a crucial role in affecting the resilient strain generated on the road pavement while in service, and *E_r_* = Resilient modulus of the asphalt concrete.

With variable changes in the ground-water table (depth to pavement surface) and capillary rise resulting in increasing saturation of the sub-grade Resilient Modulus, the pavement sub-grade becomes weak and gradually deteriorates as a result of bottom-up failure (Mode II). Equation (4) gives the rate of failure (deflection of the sub-grade).
(4)$$\Delta_{psoil}=\beta_{s1}k_{s1}\varepsilon_{v}h_{soil}\left(\frac{\varepsilon_{0}}{\varepsilon_{r}}\right)e^{-{\left(\frac{\rho }{n}\right)}^{\beta }}$$
where: $\Delta_{p\left(soil\right)}$= Permanent or plastic deformation for the layer/sublayer; *n* = Number of axle-load repetitions; $\varepsilon_{0}$ = $\varepsilon_{r}$ = Resilient strain imposed in laboratory test to obtain material properties; $\varepsilon_{v}$ = Average vertical resilient or elastic strain in the layer/sublayer and calculated by the structural response model; *h_soil_* = thickness of the unbound layer/sublayer; *K_si_* = global calibration coefficients; where *k*_*s*1_ varies from 1.673 for granular materials and 1.35 for fine-grained materials; *e_s_*_1_ = local calibration constant for the rutting in the unbound layers. The local calibration constant was set to 1.0 for the global calibration effort; *b_s_*_1_ = local or mixture field calibration constant and was set to 1.0.

Several themes, explained below, were identified for analysis in this study, including: traffic loading; material characteristics and pavement geometry; environmental and climatic conditions.

#### 2.2.1. Traffic Loading

Owing to the complexity of traffic loading, the behaviour of the underlying layers depends on the nature of the applied loading, the type of axle load, the magnitude of loading and the mode of loading either static, dynamic or cyclic. Furthermore, the associated pavement distress witnessed/experienced is influenced by the pavement layer thickness.

#### 2.2.2. Material Characterisation and Pavement Geometry

Linear Material Elastic strain response increases with increasing stress value. In its simplest form, the modulus of elasticity of a material is dependent on its stress to strain ratio. In general, the stiffness of a material is a function of its Resilient Modulus. Since each layer comprises different material geometry, the stiffness varies from top to bottom, with the topmost layer having the highest stiffness value and the preceding, inter-layers having decreasing stiffness. The layer component of the pavement surface is composed of a heterogeneous mixture of different materials. Although most paving materials are not elastic, as the materials experience some permanent deformation after each load repetition, if the load is small compared with the strength of the material, and is repeated often, the deformation under each individual load repetition is almost recoverable and is, therefore, considered to be elastic [33]. Figure 5 shows the strains under cyclic loading.

The Resilient Modulus is the general categorisation for stiffness, based on the recoverable strains encountered under repeated loading. Usually, the Resilient Modulus for each intermediate layer is defined by Equation (5):σ_ij_ = D*_ijkl_* ε*_kl_*(5)
where: σ*_ij_* is the stress tensor; D*_ijkl_* are the elastic constants; ε*_kl_* is the elastic stress tensor.

In a 2D axisymmetric analysis, using cylindrical co-ordinates, the elastic stress-strain relationship is expressed by Equation (6) [34,35]:(6)$$\left[\begin{array}{c} \sigma_{r}\\ \sigma_{\theta }\\ \sigma_{z}\\ \tau_{rz} \end{array}\right]=\frac{M_{r}}{\left(1+v\right)\left(1-2v\right)}\;\left[\begin{array}{cccc} 1-v & v & v & 0\\ v & 1-v & v & 0\\ v & v & 1-v & 0\\ 0 & 0 & 0 & \frac{1-2v}{2} \end{array}\right]\;\left[\begin{array}{c} \varepsilon_{r}\\ \varepsilon_{\theta }\\ \varepsilon_{z}\\ \gamma_{rz} \end{array}\right]$$

However, it is necessary to point out that there are certain factors that affect the resilient response of the pavement granular layers. Lekarp et al. [37,38] noted that moisture, stress level, density, grading, and maximum grain size of the material were affected by the aggregate type, shape, and angularity. Since these parameters are dynamic in nature, having a non-linear response, a strategic response must be generated to counter the effect on the stress response mode [39]. This can be achieved by improving the stiffness index of each independent layer under service loads. The maximum grain size distribution of the aggregate used, as well as the percentage of binder content greatly influence the stiffness index. The standards for bituminous mix, as specified by the South African Pavement Design Guide [36], should be adhered to, as well as the standards provided by Huang and Dong et al. [5,40].

#### 2.2.3. Environmental and Climatic Conditions

In designing pavement structures, the biggest task is to take into consideration the different climate and environmental conditions within a given area [39]. However, a better understanding of the meteorological and climate conditions must be noted, especially fluctuating changes in precipitation and increased number of cloudbursts, sudden decrease in precipitation over certain areas and reduced or warmer temperatures that have caused significant changes to the cryosphere. Values for reduction in the snow cap in areas within the northern hemisphere and increased loading caused by the formation of a thin ice layer in permafrost regions resulting in soil instability and drainage problems must be taken into account in the design variables [38]. The implementation of smart sensor devices with regards to changing climate conditions will assist in collecting more precise load variation data for analysis [41,42,43,44].

#### 2.2.4. Pavement Evaluation Matrix

The deformation of pavement constitutes the action and reaction of stress and strain development within the pavement structural matrix [41,44]. The energy-based constitutive model for the pavement matrix is based on the plastic strain energy model and this is represented by a unique relationship between the modified plastic strain energy and the stress parameter [45,46]. The evaluation of pavement using smart models in computing methods (data engine, programming, instrumentation and monitoring devices) gives rise to an evaluation matrix as defined by major constitutive Pavement Quality Indices to determine the Safety Index as well as the prediction of failure during service load [47]. Figure 6 contains a sketch showing the layout for the experimental design using smart sensor probes in real-time. The data collected via the sensors were stored on a cloud system. This was performed using an Amazon Web Service (AWS) programme written specifically for this study in synchrony with the Arduino Block diagram shown in Figure 6.

The data collected were then mined and sorted so that true pavement response information could be analysed further. There could be noise variations in the data being collected, which could result from pipe leakage or unforeseen circumstances. Although, in the event of pipe leakages, the analysis would present a consistently increasing moisture content that would require immediate action. The data being mined were then analysed using a Distinctive Mesh Control analysis in Abaqus CAE (Finite Element Analysis) [48,49,50]. The output data after analysis are presented in Section 3.

For this study, a number of Pavement Quality Indices have been presented in Table 1 to take into account the respective measures to counter pavement deformations as they occurred. Depending on the cause of pavement failure, a responsive and smart remedial approach was required to counter the damage that will occur resulting from the distress mode experienced by the pavement [51,52,53]. Figure 7 provides the backend domain repository statistics where the sensor data is collected and stored within a Sigfox network. The Sigfox network is a web service system used to collect and store different ranges of data within a registered secured domain https://build.sigfox.com/sigfox (accessed on 26 April 2021). Figure 8 and Figure 9 provides data collected by the sensors and brought forward to the front end of MISRA (Moisture Instrumentation Sensor for Road Analysis) software platform. This platform is a Java based program analysis website hosted on the Amazon Web Service system. The data collected from the sensors is stored in a repository domain (https://backend.sigfox.com/device/list, accessed on 1 May 2021). The data from the sensor (https://backend.sigfox.com/device/list) is linked to the front end www.misraanalysis.co.za (accessed on 1 May 2021) using a compiler program and presented in Figure 8 and Figure 9.

## 3. Analysis and Modelling

Pavement Structural Adequacy (PSA) is the primary index in defining the resistance of pavement to design loads, besides Riding Quality Index and Visual Quality Index. PSA defines the stiffness index of pavement with regards to the Resilient Modulus. Although certain models have been defined to determine the Resilient Modulus according to the M-E Pavement Design Guide 2004 [34], it must be noted that a Work Hardening Energy Constitutive Model is needed to analyse the structural adequacy. Since strain increment (axial, lateral, shear, and volumetric) during loading is dependent on the magnitude of time-based stress, either static or dynamic, a special energy function modelling exists, using elasto-plastic models to generate yield functions in form (Mohr-Coulomb and Druker-Prager Models). Finite element analysis using Abaqus http://130.149.89.49:2080/v6.13/books/usi/default.htm (accessed on 26 April 2021); provides the Von-Mises stress and strain values which indicate the resistance to deformation. It is necessary to note that structural adequacy is dependent on the matrix of the asphalt-concrete mix. This matrix comprises the distribution of the aggregate size as well as the summation of the composite structural stiffness of the pavement [45]. A simple model for pavement failure analysis using axisymmetric modelling has been presented with the application of a Drucker-Prager yield function model in the Abaqus CAE model using Distinctive Mesh Control https://abaqus-docs.mit.edu/2017/English/SIMACAEMATRefMap/simamat-c-druckerprager.htm (accessed on 26 April 2021).

### Discrete Analysis

The results of the Pavement Instrumentation Humidity values and Pavement Instrumentation Temperature values obtained from sensor probes are presented in Figure 8 and Figure 9 respectively. Furthermore, the results of the discrete analysis to determine the Griffith Fracture Energy and pavement modulus are presented in Table 2. 

Figure 8 indicates that the humidity/moisture ingress data collected over the period of time for the road section clearly showed failure of the pavement prior to the end of its design life. The trigger points with the highest humidity values indicated that there was a sudden increase in the moisture content resulting from increasing rainfall within the period of instrumentation (November to February 2019). The increasing humidity value gave an indication of the moisture saturation content of the pavement. The higher humidity value (>40 to 170) from experimental laboratory tests, using the sensor device, showed that humidity values for the pavement had exceeded the required pavement design humidity requirement. Figure 9 shows the temperature data collected from the pavement during the time of instrumentation. The temperature values collected from the pavement ranging above 25 degrees Centigrade indicated a threat to the pavement, resulting in wrapping and rutting of the asphalt-concrete pavement layer under service load.

Discrete analysis was performed to determine the Griffith Fracture Energy and pavement stiffness modulus. The main feature regarding the use of a discrete analysis method is the ability to generate fracture failure response in any direction by selectively breaking the bonds between the individual discrete elements of the pavement structure through the ingress of moisture into the pavement [45,46]. The development of fracture within a continuous medium in discrete analysis, is a function of its strength properties in relation to brittle or ductile materials. This is a usual occurrence in flexible pavement having a viscoelastic behavior [52,53,54]. The onset of failure is further developed from the crack opening within the pavement structure. The onset of cracking at the midpoint of the pavement element is governed by a damage model that is usually a concept in discrete analysis. This damage is caused by reduced stiffness resulting from moisture ingress into the pavement structure [54,55]. For the purpose of this study, the use of a moisture sensor is provided to collect the data regarding moisture variation and temperature variation over the specified period as presented in Table 2. A detailed analysis indicating the magnitude of data collected and evaluation performed has been presented. From Table 2, it was observed that decreasing temperature increases the Griffith Fracture Energy. The Griffith Fracture Energy [54,55,56], which is also related to the pavement stiffness modulus, behaved in such a way that increasing Griffith Fracture Energy increased the stiffness or Resilient Modulus of the asphalt-concrete over time. Temperature gradient above 25 degrees, with further temperature gradient variations up to 35 degrees, resulted in decreasing Griffith Fracture Energy of the pavement damage over a period of time. Thus, it was necessary to monitor the temperature variation of the pavement over time.

The results indicated the efficiency of collecting real-time data from pavements using sensor probes (micro-controllers, and data memory card powered by a solar-panel system) over a period of time. Furthermore, the results (shown in Figure 10 and Figure 11, and further results in Appendix B) served as input data for high-level, real-time analysis, and the results with regards to changes in temperature against fracture energy have been presented. Since pavement behaves like a visco-elastic material [50,51,52,53,54], performance of the pavement tested was influenced significantly by variations in moisture and temperature gradient [57,58]. The modulus of the asphalt was greatly affected, hence the reduction in strength over a period of time. The data from the instrumentation process were analysed using data mining tools in regression models. Table 2 shows the data collected and the pavement analysis report over a given period of time. The asphalt-concrete modulus and the Griffith Fracture Energy for pavement failure were computed. The Temperature and Humidity data files to support the findings of this study have been included in the Appendix A, as well as on the web domain: www.misraanalyst.co.za (accessed on 26 April 2021) Access to this domain is by request as this is Intellectual Property under review and Non-Disclosure Confidentiality is of paramount importance.

The cost implication of any pavement structure determines its ease of construction and maintenance. If an appropriate pavement response model exists, there will be a more efficient pavement management system in the event of pavement damage during its service life. Figure 10 and Figure 11 show the relationship between pavement temperature and the Asphalt-concrete Fracture Energy from the sensor device. The relationship indicated that an increase in temperature reduces the Asphalt-concrete Fracture Energy. Fracture Energy is the ability of a material to resist failure by possessing internal resistance as a result of its material property or stiffness coefficient. The resulting graph shows that, at lower temperatures, the asphalt-concrete was stronger and could sustain more loads, thereby increasing the Pavement Quality Index.

## 4. Discussion

Considering pavement failure with regards to rutting and cracking variables [6], Figure 12 and Figure 13 present a new approach to assessing pavement failure by considering deformation equations other than fatigue rutting and cracking [52,53]. This is calculated in terms of: the relationship between the pavement’s Fracture Energy and pavement resilient modulus, which is related to the effects of the asphalt-concrete temperature and moisture. The results showed that increasing asphalt-concrete temperature reduced the strength of the unbound granular base. This further resulted to reduction in the elastic modulus of the sub-base layer and reduced the strength of the pavement as indicated in Figure 10, Figure 11, Figure 12 and Figure 13, leading to the formation of bottom-up cracks and cracking failure. The vertical compressive strain (E11) at the sub-base layer, at 20% moisture content, was 69.57 × 10^−4^ which increased to 140.8 × 10^−4^ at 60% moisture content [56]. The horizontal deformation (E22) reduced, indicating that the material was experiencing work hardening and no further stress could result in any significant damage. The damage remained at a constant value of 96.8 × 10^−4^ at 60% saturation. Consequently, the performance of the pavement was affected by temperature gradient. Increasing temperature gradient resulted in reduction in stiffness of the asphalt layer. In tropical regions, this can result in immediate rutting failure of the asphalt layer which, over time, leads to the formation of top-down cracks and potholes with increasing moisture content. Advanced computations in pavement engineering, which entail the creation of the semantics of the pavement domain, during its design life, through artificial intelligence and high-level, expert system shell analysis in JAVA programming, has been proposed for a future study. This model synchronises the Pavement Management System with a web-based data architecture, using smart technology for pavement analysis [56,57,58].

## 5. Conclusions

The introduction of a responsive pavement brings about efficiency in the pavement life as well as the performance of the pavement while in service. This serves as a premise towards the development of a Pavement Quality Index for the test section considered. Since the use of the distress catalogue from previous research is time consuming and substantial in analysis, a simplified and efficient method to predict and analyse road failure, which is able to provide the stiffness ratio of the pavement as well as the fracture energy of the pavement at any given point in time, has been presented. When compared with standard methods of analysis such as fatigue fracture and fatigue cracking methods as laid out in multi layered elastic design [5,10,59], the development of pavement delamination using sensors which take readings in real times helps to be more efficient since engineering judgement regarding maintenance actions can be decided within a reduced turnaround time. From the sensor data collected, the variable parameters observed from the sensors, such as reduced temperature gradient, increase the Fracture Energy within the pavement structure. This phenomenon makes the pavement stiffness high and prone to failure but, at very low temperatures, a compromise is reached and the strength is breached, resulting in a brittle material (glass). Although the failure is not visible at the onset of crack propagation, continual exposure to increasing temperatures, as well as increasing moisture content, will lead to failure of the pavement before the design life is reached. There is also a strong linear relationship between Pavement Fracture Energy and the Pavement Resilient Modulus.

For the purpose of the study conducted, the problem identified in the pavement was caused by moisture and temperature damage resulting in premature failure prior to the end of the design life. The objective of the study was to propose a Pavement Quality Index rating for Smart Road Infrastructure using smart pavement-sensors to collect and analyse pavement temperature and moisture saturation content to assess the quality and performance of the pavement as well as to predict pavement failure within any specific time-frame, in real-time, using constitutive models within the Mechanistic-Empirical Pavement Design Guide synchronised with a Pavement Management System model. The results were analysed using constitutive models within the Mechanistic-Empirical Pavement Design Guide. A Pavement Quality Index Smart Pavement-Management System (PQISMS) was proposed that was designed to produce a smart responsive report sheet in real-time, considering moisture and temperature gradient variations in synchrony with a web-based pavement deformation architecture model. The major findings of the study outline a Pavement Quality Index rating, using a smart sensor device for monitoring for pavement structural health in real-time to analyse pavement responses under changing environmental conditions. The limitation of the study was that the use of solar panels to charge the battery embedded in the Box (micro-processor) resulted in further cutting of the pavement surface to make it possible to connect the wires to the solar panel that is usually mounted on the street lights. Further study on smart pavement-instrumentation will require a sustainable approach to charging the batteries, either using a kinetic-energy charging system or via connection to underground fibre cables. A novel, smart sensor device to monitor pavement condition was generated in the course of the study.

## 6. Patents

A novel, road-condition sensor prototype was developed in the course of the study and a patent application was granted fully and secured with the South African Patent Attorneys (Ref: Adams and Adams). Patent Reference Number: P86886ZP00 VIP/jbg.

## Figures and Tables

**Figure 1 sensors-21-04231-f001:**
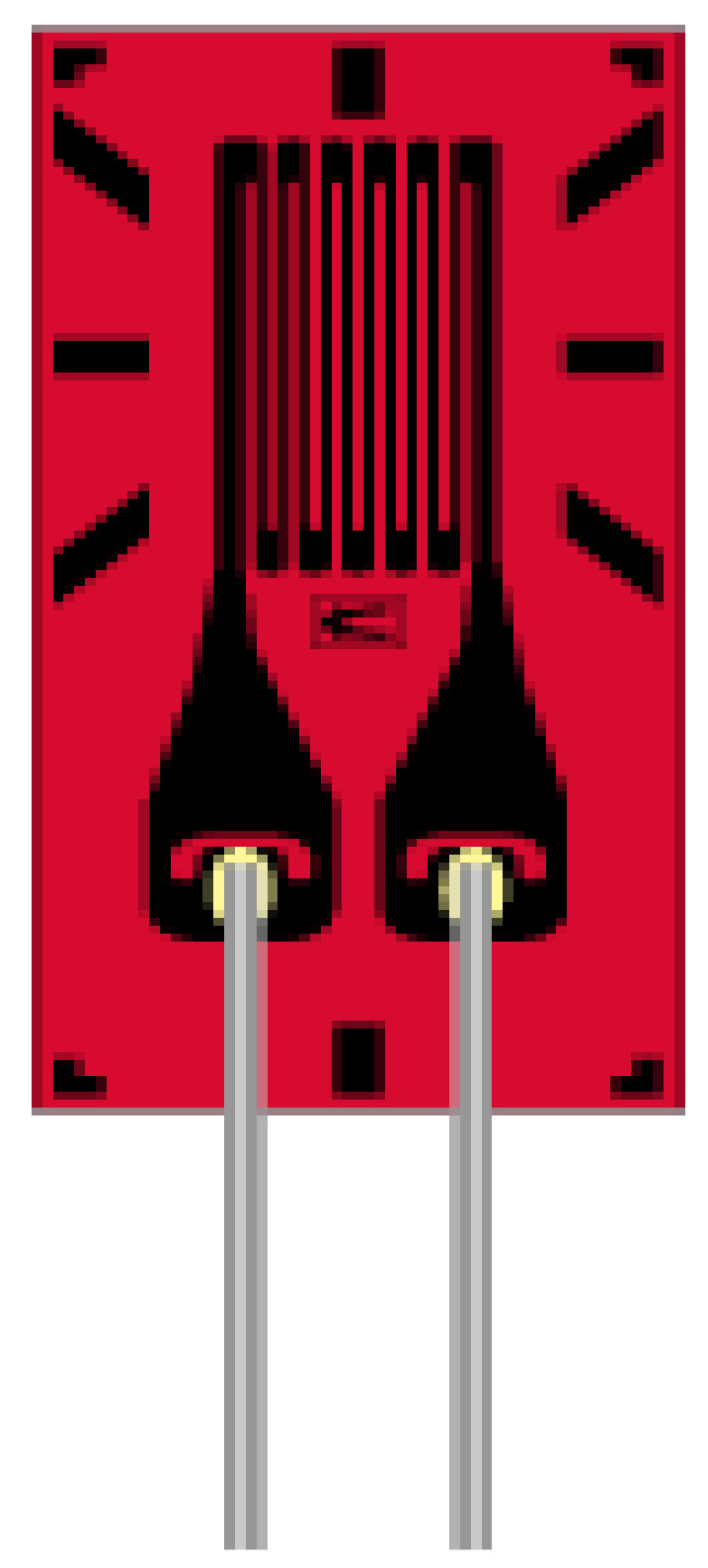
Pictorial representation of H—Gauge manufactured by Kyowa.

**Figure 2 sensors-21-04231-f002:**
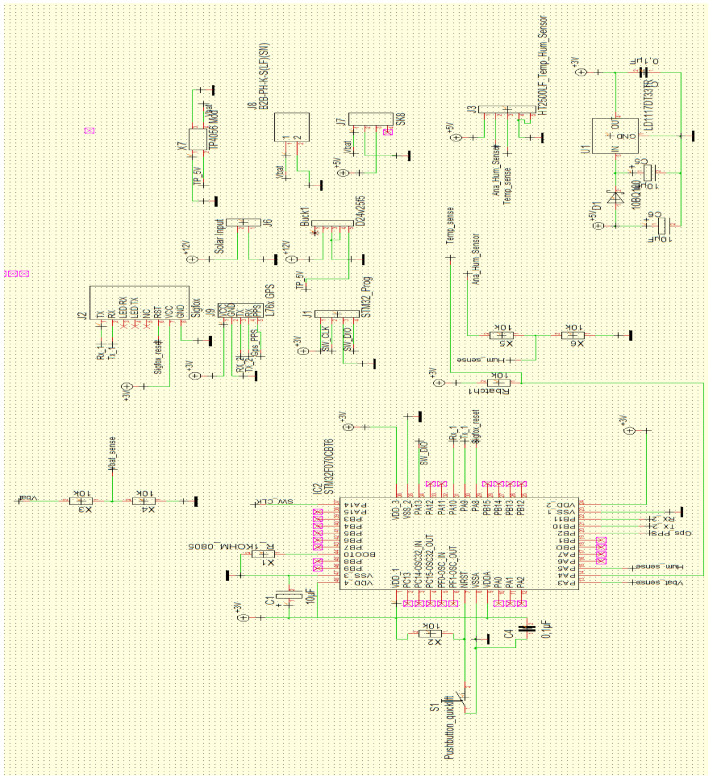
Proposed Arduino Block Diagram for Determining Moisture and Temperature Values.

**Figure 3 sensors-21-04231-f003:**
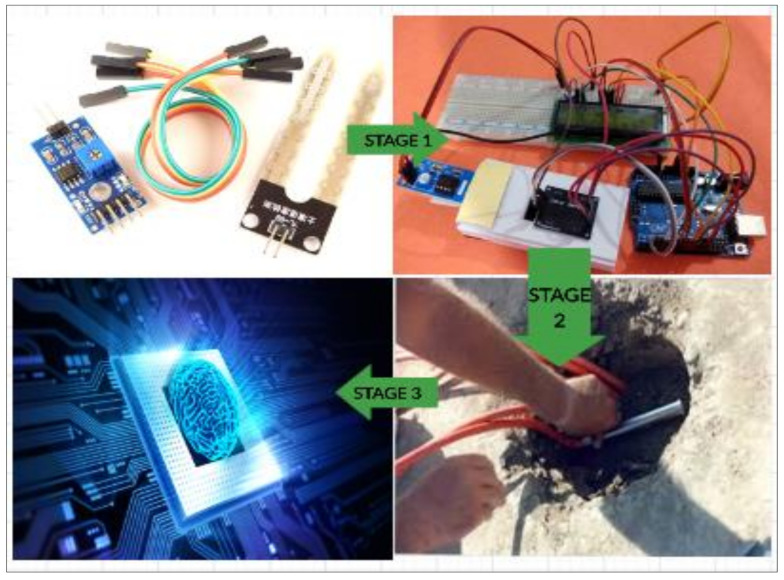
Proposed Technology for Moisture Instrumentation of Pavement Sub-Grade Layer.

**Figure 4 sensors-21-04231-f004:**
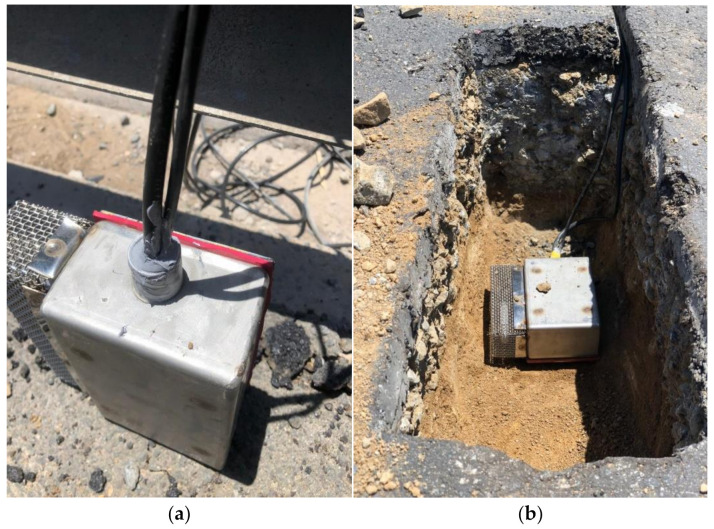
Smart Instrumentation of Pavement using Moisture Instrumentation Sensor for Pavement Analysis. (**a**) A top view of the sensor at the instrumentation site; (**b**) Pictorial view of the sensor embedded into the road before refill and compaction

**Figure 5 sensors-21-04231-f005:**
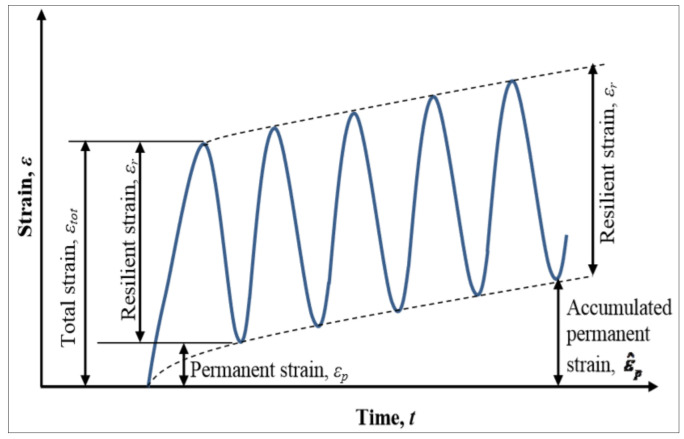
Strains under Cyclic Loading [37].

**Figure 6 sensors-21-04231-f006:**
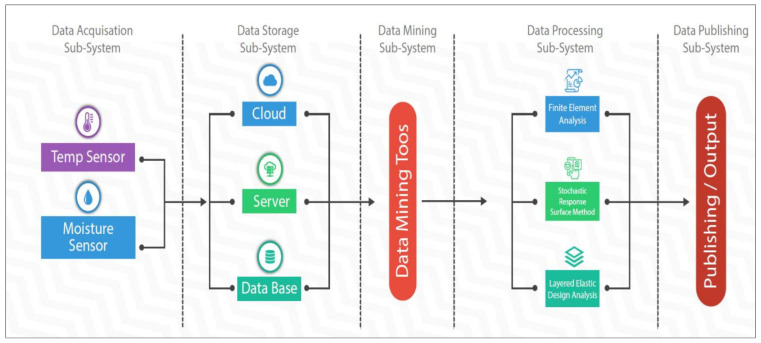
Experimental Layout for Smart Pavement-Instrumentation and Modelling.

**Figure 7 sensors-21-04231-f007:**
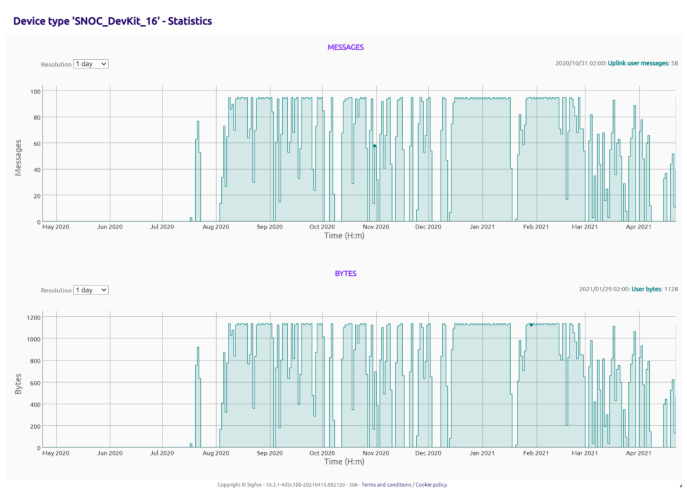
Pavement Instrumentation Humidity Values Obtained from Sensor Probes.

**Figure 8 sensors-21-04231-f008:**
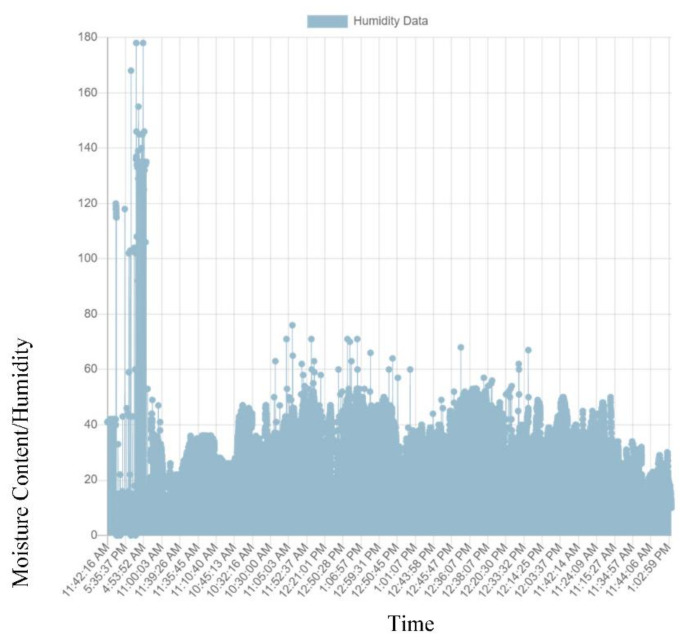
Pavement Instrumentation Humidity Values Obtained from Sensor Probes.

**Figure 9 sensors-21-04231-f009:**
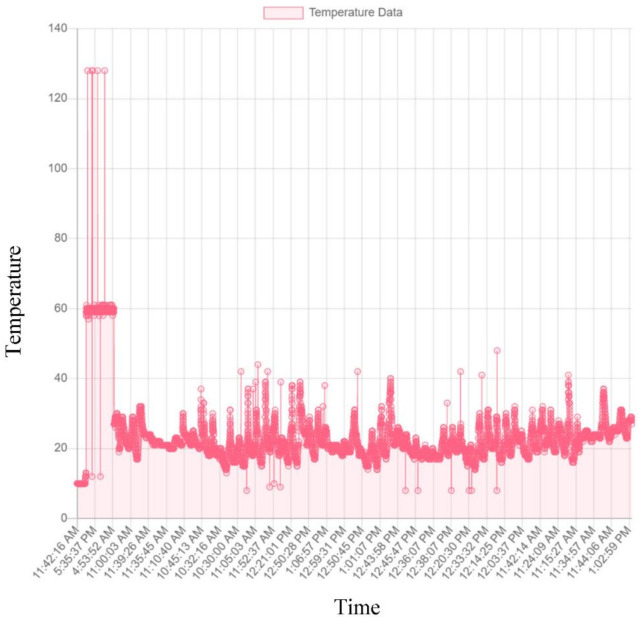
Pavement Instrumentation Temperature Values Obtained from Sensor Probes.

**Figure 10 sensors-21-04231-f010:**
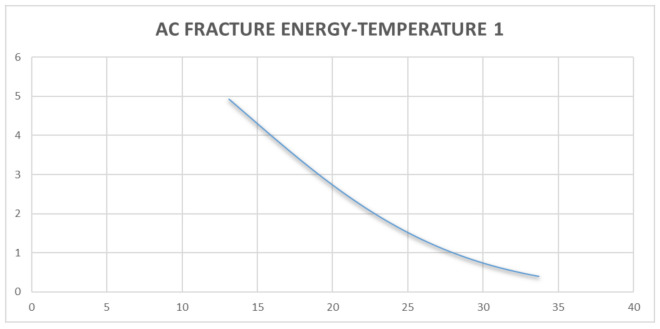
Asphalt-Concrete Fracture Energy against Temperature Sensor 1.

**Figure 11 sensors-21-04231-f011:**
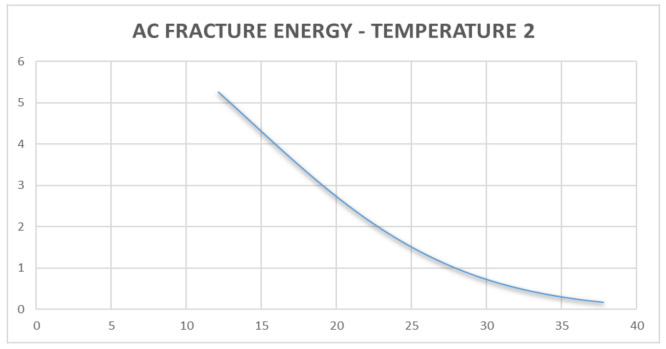
Asphalt-Concrete Fracture Energy against Temperature Sensor 2.

**Figure 12 sensors-21-04231-f012:**
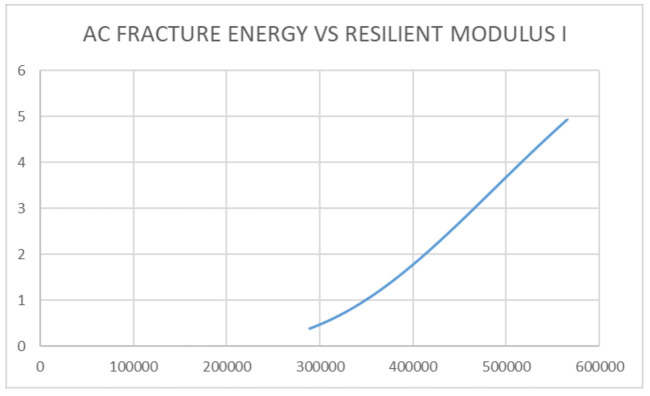
Relationship between AC Fracture Energy and Resilient Modulus E1.

**Figure 13 sensors-21-04231-f013:**
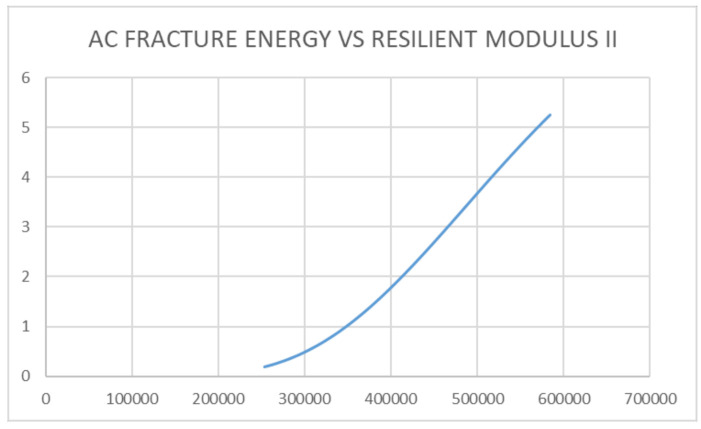
Relationship between AC Fracture Energy and Resilient Modulus E2.

**Table 1 sensors-21-04231-t001:** Pavement failure modes indicators.

Pavement Distress	Pavement Deterioration	Counter Measure
Pavement Cracking	Traffic Loading	Appropriate pavement thickness design; Diversion of traffic where necessary
Pavement Distortion	Environmental/Climatic Factors	Appropriate camber requirement; Appropriate grade or slope design
Pavement Disintegration	Drainage Effect	Provision of side ditches, verges or side drains; Appropriate slope and camber design
Skidding Resistance	Material Quality	Appropriate selection of material requirement and appropriate laboratory checks for quality and standards control
Surfacing/Roughness	Utility/Service Cuts	Appropriate re-sealing where necessary, otherwise avoid cutting pavement for services; Use under-pavement bore for fibre connection cables and utility services
Mode I Failure	Construction Defects	Appropriate supervision and compaction requirement

**Table 2 sensors-21-04231-t002:** Relationship between Griffith Fracture Energy and Pavement Resilient Modulus.

Time	T1	T2	GT (T1)	GT (T2)	E (T1)	E (T2)
10	23.87	24.1	1.751691	1.710227	398,204.4	395,738.2
20	24.62	24.8	1.59182	1.565145	388,558.1	386,910
30	25.06	25	1.502769	1.514706	383,008.1	383,760.3
40	24.94	24.9	1.526708	1.540793	384,513.9	385,394.9
50	25.12	25.1	1.490896	1.502769	382,257.5	383,008.1
60	25.37	25.2	1.442125	1.477127	379,145.6	381,383.6
70	25.75	25.6	1.370139	1.394475	374,463.9	376,059
80	26.06	25.9	1.31332	1.335112	370,687.6	372,144.8
90	26.12	25.9	1.302519	1.335112	369,961	372,144.8
100	26.19	26	1.289999	1.324184	369,115.3	371,415.5
110	26.37	26.3	1.2582	1.279336	366,949.2	368,391.8
120	26.69	26.6	1.203068	1.22525	363,129.9	364,676.7
130	26.94	26.8	1.161233	1.182852	360,173.7	361,707.9
140	27.12	26.8	1.131777	1.192929	358,060.2	362,418.2
150	27.12	26.8	1.131777	1.192929	358,060.2	362,418.2
160	27.12	26.7	1.131777	1.203068	358,060.2	363,129.9
170	27.06	26.6	1.141534	1.214976	358,763.3	363,962
180	27.25	26.9	1.110848	1.161233	356,541.4	360,173.7
190	27.5	27.1	1.071404	1.141534	353,638.9	358,763.3
201	27.75	27.1	1.03301	1.131777	350,760	358,060.2

Legend: T1: refers to temperature on sensor probe 1; T2: refers to temperature gradient on sensor probe 1; GT: Griffith Fracture Energy on sensor probes readings; E: Change in Resilient Modulus of the asphalt concrete.

## Data Availability

Not applicable.

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
