# Peer review of "Pavement Quality Index Rating Strategy Using Fracture Energy Analysis for Implementing Smart Road Infrastructure"

_sensors, 2021, doi:10.3390/s21124231_

Round 1

Reviewer 1 Report

There are three abbreviations IRI, PCI, and ASTM, that need to be presented in an extensive form the first time they appear in the text.

The presentation of H-gauges manufactured by Kyowa Gauges with the aid of a schematic should be interesting.

The authors should rewrite the description of the criterion of maximum aggregation size because it is not understandable.

The authors must explain why they use three Temperature and Soil Moisture sensors and how they transmit the data to the Amazon Web Service (AWS) because Figure 1 shows the Arduino writing in an SD Card.

Some information or internet web site of Abaqus software should be valuable because some readers would not know it.

There is a lack of information about the discrete analysis that determined the Griffith Fracture Energy and Pavement Resilient Modulus in Table 2. Table 2 also needs a legend or a paragraph to describe its columns.

On line 566 and 567 the text mention Figure 7 and Figure 8, they should be Figure 8 and Figure 9, respectively. The schematic shown in Figure 1 represents three and not two sensors in the experiment.

The authors must plot a graph of the linear relationship between Pavement Fracture Energy and Pavement Resilient Modulus.

Author Response

REVIEWER 1

Comments and Suggestions for Authors

Comment: There are three abbreviations IRI, PCI, and ASTM, that need to be presented in an extensive form the first time they appear in the text.

Response: This has been taken care of as indicated (Line 103, 255 and 275 and 276)

The presentation of H-gauges manufactured by Kyowa Gauges with the aid of a schematic should be interesting. (Line 193)

Response: Noted (Line 193)

Comment: The authors should rewrite the description of the criterion of maximum aggregation size because it is not understandable.

For pavement application, the working principle is based on the maximum aggregate size of the pavement mixture matrix, usually taken to be 25.4mm. The criterion for maximum aggregate size method was developed by [23] which relates the level of flexibility resulting from movement of the aggregate molecules within the pavement structure (meso structure) [23]. The maximum aggregate size takes account of the parameters of the multi-phase properties considered within the pavement material such as; shape, size, gradation and distribution of the aggregate particles. The interaction or friction developed between the maximum aggregate size and the bituminous binder defines the resultant strain developed within the pavement structure [23; 24; 25]. This change in length, or deflection caused results into induced strain variations occurring within the pavement structure measures using strain gauge.

Please refers to lines 175 - 182

It is noted that the meso-scale structure is the principle behind the maximum aggregate size but is not detailed in the text. However, a reference is made below for the clarification of the reviewer.

In meso-scale structures, concrete is considered as a multi-phase material either double phase aggregate and mortar, or three-phase (mortar, aggregates, and interfaces). The complex problems of the cracking analysis of concrete was a big challenge. Many numerical models adopted for the fracture mechanic analysis of concrete such as qusai-brittle materials in the past few decades. These numerical models also tried to interpret the energy dissipation that occur during the fracture process zone. [23].

Comment: The authors must explain why they use three Temperature and Soil Moisture sensors and how they transmit the data to the Amazon Web Service (AWS) because Figure 1 shows the Arduino writing in an SD Card.

Response: The Arduino block diagram in Figure 2 shows the proposed method for determining the characteristic variables that change over time during the life-cycle of the pavement. The block diagram shows the microprocessor connection between the sensor and the data storage system provided in synchrony with the Amazon Web Service. The Figure 6 further provides the synchrony between the Arduino sensor (temperature and moisture/humidity) system and the AWS algorithm system. In the synchronized design between AWS and Arduino, one sensor device directly connects to the web service domain. The output provides the use of two sensors differently installed to collect relevant data which the mean value can be derived for argument sake as well as detecting errors should there be failure of the sensor or inability to send data in scenarios of low battery voltage or fully drained battery power.

Response: (Line 296 to 304). Figure 2 shows the proposed system for implementing smart road-management infrastructure using moisture and temperature sensors to measure the condition of the road at any given time. The Finite Element Method of analysis and Multi-Layer Elastic Design of pavement are still in use but do not give real-time solutions to determine a Pavement Quality Index. So, a system synchronizing Arduino and Amazon Web Service (AWS) was used in this study.  The Arduino block diagram in Figure 2 shows the proposed method for determining the characteristic variables that change over time during the life-cycle of the pavement. The block diagram shows the microprocessor connection between the sensor and the data storage system provided in synchrony with the Amazon Web Service. The Figure 6 further provides the synchrony between the Arduino sensor (temperature and moisture/humidity) system and the AWS algorithm system.

Please refer to the supplementary file.

Comment: Some information or internet web site of Abaqus software should be valuable because some readers would not know it.

Response: Noted and duly resolved (Line 527 to Line 535)

Comment: There is a lack of information about the discrete analysis that determined the Griffith Fracture Energy and Pavement Resilient Modulus in Table 2. Table 2 also needs a legend or a paragraph to describe its columns.

Response: Noted and resolved (Line 566 to 575)

The main feature regarding the use of a discrete analysis method is the ability to generate fracture failure response in any direction by selectively breaking the bonds between the individual discrete elements of the pavement structure through the ingress of moisture into the pavement [45; 46]. The development of fracture within a continuous medium in discrete analysis, is a function of its strength properties in relation to brittle or ductile materials.  This is a usual occurrence in flexible pavement having a viscoelastic behavior [43; 46]. The onset of failure is further developed from the crack opening within the pavement structure. The onset of cracking at the midpoint of the pavement element is governed by a damage model usually a concept in discrete analysis. This damage is caused by reduced stiffness resulting from moisture ingress into the pavement structure [47; 48].

Also, a new reference is added as follows:

Zárate, F., Oñate, E. A simple FEM–DEM technique for fracture prediction in materials and structures. Comp. Part. Mech. 2, 301–314 (2015). https://doi.org/10.1007/s40571-015-0067-2

Comment: On line 556 and 559 the text mention Figure 7 and Figure 8, they should be Figure 8 and Figure 9, respectively. The schematic shown in Figure 1 represents three and not two sensors in the experiment.

The Arduino block diagram in Figure 2 shows the proposed method for determining the characteristic variables that change over time during the life-cycle of the pavement. The block diagram shows the microprocessor connection between the sensor and the data storage system provided in synchrony with the Amazon Web Service. The Figure 6 further provides the synchrony between the Arduino sensor (temperature and moisture/humidity) system and the AWS algorithm system. In the synchronized design between AWS and Arduino, one sensor device directly connects to the web service domain. The output provides the use of two sensors differently installed to collect relevant data which the mean value can be derived for argument sake as well as detecting errors should there be failure of the sensor or inability to send data in scenarios of low battery voltage or fully drained battery power.

Response: Noted and resolved (Line 566 and 567)

Comment: The authors must plot a graph of the linear relationship between Pavement Fracture Energy and Pavement Resilient Modulus.

Response: Noted and provided

Please refer to….. (Line 633 and 637)

Reviewer 2 Report

In this article, the development of a pavement management system at the network level was examined to contribute towards a framework for evaluating a Pavement Quality Index, and service life capacity. Environmental surface response models in the form of temperature and moisture variations within the pavement were applied, using sensor  devices connected to a data cloud system to carry out mathematical analysis using a distinctive mesh analysis deformation model.  The results indicated variation in the Resilient Modulus of thepavement, with increasing moisture content.

The paper is well written, however, the authors need to address the following concerns:

1. The quality of figures need to be improved, specifically figure 1, 4 (fonts and some notations in the figure),  and figure 6.

2.  Scientific Soundness (Theoretical contribution of this paper) 
 The authors claim "Since the use of the distress catalogue from previous research is time consuming and substantial in analysis, 
a simplified and efficient method to predict and analyze road failure, which is able to provide the stiffness  ratio of the pavement as well as the fracture energy of the pavement at any given point in time, has been presented"
But though we see some mathematical equations, we do not see any discussion of their proposed methods in regards of 
existing methods. In addition, their new approach was not simulated theoretically so that we can see its theoretical merit before 
assessing it in terms of implementation"
There is a confusing aspect of reporting a project's results and demonstrating a research achievement. The title talks about strategy
or framework but within the paper, they mentioned method and technique. I think the title should mention ..." framework/ or strategy using this technique for smart road infrastructure". And the authors can discuss the output of their technique. The discussion section could have a comparison 
table of their technique and existing techniques in regards to some features/metrics

Author Response

REVIEWER 2

Comments and Suggestions for Authors

In this article, the development of a pavement management system at the network level was examined to contribute towards a framework for evaluating a Pavement Quality Index, and service life capacity. Environmental surface response models in the form of temperature and moisture variations within the pavement were applied, using sensor devices connected to a data cloud system to carry out mathematical analysis using a distinctive mesh analysis deformation model.  The results indicated variation in the Resilient Modulus of the pavement, with increasing moisture content.

The paper is well written, however, the authors need to address the following concerns:

Comment :The quality of figures need to be improved, specifically figure 1, 4 (fonts and some notations in the figure),  and figure 6.

Response: Noted and resolved. The figures from 6 and 7 were developed from the AWS domain, a clearer Figure is provided as seen in… (Line 556 and 559)

Comment: Scientific Soundness (Theoretical contribution of this paper) 
The authors claim "Since the use of the distress catalogue from previous research is time consuming and substantial in analysis, a simplified and efficient method to predict and analyze road failure, which is able to provide the stiffness  ratio of the pavement as well as the fracture energy of the pavement at any given point in time, has been presented" But though we see some mathematical equations, we do not see any discussion of their proposed methods in regards of 
existing methods.

Response: (Line 664 tp 668) a simplified and efficient method to predict and analyse road failure, which is able to provide the stiffness ratio of the pavement as well as the fracture energy of the pavement at any given point in time, has been presented. When compared with standard methods of analysis such as fatigue fracture and fatigue cracking methods as laid out in multi layered elastic design [5; 10; 59]

Comment: In addition, their new approach was not simulated theoretically so that we can see its theoretical merit before assessing it in terms of implementation"

Response: A part of the simulation was made using the algorithm presented in the supplementary file. The simulation is RUN using an online web domain (https://www.onlinegdb.com/online_c++_compiler) The code was written using a C++ language and compiled using Visual basic studio. The algorithm is attached as a supplementary file

The analysis document is presented in the supplementary file as a separate runtime algorithm which was not included in the manuscript. The simulation is presented in form of combination of equations which was filed in a patent document provided in a supplementary file

Simulations in the appendix are in a supplementary file alongside the coding file and analysis file

Comment: There is a confusing aspect of reporting a project's results and demonstrating a research achievement. The title talks about strategy or framework but within the paper, they mentioned method and technique. I think the title should mention ..." framework/ or strategy using this technique for smart road infrastructure". And the authors can discuss the output of their technique. The discussion section could have a comparison table of their technique and existing techniques in regards to some features/metrics

Response: The title is revised. Pavement Quality Index Rating Strategy Using Fracture Energy Analysis for Implementing Smart Road Infrastructure

Reviewer 3 Report

The topic of this manuscript as well as proposed methodology is new and interesting, but the way of presentation and mathematical models description do not meet Sensors journal requirements. 

Suggestions:

1. please double check manuscript: there are missed words, some key parameters are not described properly (e.g. in formula (3))

2. please discuss more in details the contemporary methods on Pavement Quality  control and defects detection  recently proposed in the literatures, incl. [1].

[1] Nguyen, T. H.; Nguyen, T. L. et al. Machine learning algorithms application to road defects classification . Mar 2018, Intelligent Decision Technologies. DOI: 10.3233/IDT-170323

Author Response

REVIEWER 3

Comments and Suggestions for Authors

The topic of this manuscript as well as proposed methodology is new and interesting, but the way of presentation and mathematical models description do not meet Sensors journal requirements. 

Suggestions:

Comment: please double check manuscript: there are missed words, some key parameters are not described properly (e.g. in formula (3))

Response: Resolved (Line 403 to 405)

Comment:. please discuss more in details the contemporary methods on Pavement Quality control and defects detection recently proposed in the literatures, incl. [1].

[1] Nguyen, T. H.; Nguyen, T. L. et al. Machine learning algorithms application to road defects classification. Mar 2018, Intelligent Decision Technologies. DOI: 10.3233/IDT-170323.

Response: (Line 358 to 362). This study was limited to smart road infrastructure using moisture and temperature sensor instrumentation built in Arduino system and in synchrony with computational analysis performed in Amazon Web Service. The authors will the authors did not consider machine learning algorithm. This procedure using machine learning algorithms is recommended in further study.

The researcher has recently requested for this paper (Nguyen et al) from the author. We will further consider a more advanced approach with machine learning in further research on pavement analysis. This study was limited to development of a smart sensor device to analyse pavement failure considering temperature and moisturecontent/humidity variations during the service life.

In further study, a Matlab subroutine program using combined with machine learning algorithms will be considered.

Round 2

Reviewer 2 Report

The comments and suggestions were addressed accordingly 

Reviewer 3 Report

The comments have been carefully considered.

This manuscript is a resubmission of an earlier submission. The following is a list of the peer review reports and author responses from that submission.